# Postoperative Hormone Replacement Therapy and Survival in Women with Ovarian Cancer

**DOI:** 10.3390/cancers14133090

**Published:** 2022-06-23

**Authors:** Eunjeong Ji, Kidong Kim, Banghyun Lee, Sung Ook Hwang, Hee Joong Lee, Kyungjin Lee, Minkyung Lee, Yong Beom Kim

**Affiliations:** 1Medical Research Collaborating Center, Seoul National University Bundang Hospital, 82, Gumi-ro 173 Beon-gil, Bundang-gu, Seongnam-si 13620, Korea; 99145@snubh.org; 2Department of Obstetrics and Gynecology, Seoul National University Bundang Hospital, 82, Gumi-ro 173 Beon-gil, Bundang-gu, Seongnam-si 13620, Korea; kidong.kim.md@gmail.com (K.K.); ybkimlh@snubh.org (Y.B.K.); 3Department of Obstetrics and Gynecology, Inha University Hospital, Inha University College of Medicine, 27, Inhang-ro, Sinheung-dong, Jung-gu, Incheon 22332, Korea; sohwang@inha.ac.kr (S.O.H.); kglee0413@inhauh.com (K.L.); mklee@inhauh.com (M.L.); 4Department of Obstetrics & Gynecology, College of Medicine, The Catholic University of Korea, 222, Banpo-daero, Seocho-gu, Seoul 06591, Korea; heejoong@catholic.ac.kr

**Keywords:** ovarian cancer, overall survival, postoperative hormone replacement therapy

## Abstract

**Simple Summary:**

Previous studies have reported that hormone replacement therapy (HRT) after surgery for ovarian cancer has no significant effect on survival. However, a prospective study and a randomized controlled study showed that HRT administration provided survival benefits. Therefore, this study aimed to investigate the impact of postoperative HRT on survival in women with ovarian cancer using the nationwide cohort study. This cohort study was conducted on 1784 women aged ≤60 and diagnosed with ovarian cancer. Overall survival was significantly greater for women that received HRT than for women that did not after a mean follow-up of 5.6 ± 2.9 years (85.3% vs. 76.6%, respectively). Postoperative HRT may be considered a maintenance therapy in women diagnosed with ovarian cancer at ≤60 years of age.

**Abstract:**

The effect of postoperative hormone replacement therapy (HRT) on survival in women with ovarian cancer remains unclear. This study aimed to investigate the impact of postoperative HRT on survival in women with ovarian cancer using the nationwide cohort study. Women aged ≤60 and diagnosed with ovarian cancer that received primary surgery were followed-up for 5.6 ± 2.9 years. Mean ages of women administered HRT (the HRT group; n = 263) or not administered HRT (the control group; n = 1521) were 41.5 ± 8.5 and 41.0 ± 11.4 years, respectively. After adjustment for covariables, OS was significantly greater in the HRT group (HR 0.618; 95% CI 0.414–0.922; *p* = 0.018). Kaplan–Meier curve analysis showed OS was significantly higher in the HRT group (85.3% vs. 76.6%; *p* = 0.016). The ratio of women with HRT to women without HRT increased significantly with time (restricted mean survival times for OS, *p* < 0.001). In addition, OS was significantly greater for those that received HRT for >5 years than for those that received HRT for ≤0.5 years (HR 0.234; 95% CI 0.059–0.936; *p* = 0.040). Postoperative HRT improved survival among women with ovarian cancer. The impact of HRT on survival increased with time and treatment duration.

## 1. Introduction

Ovarian cancer is the most common cause of death among women with gynecologic cancers [1,2]. In the United States, an estimated 13,940 deaths were attributed to ovarian cancer in 2020 [1]. Many women with ovarian cancer become menopausal after surgical treatment [2]. Hormone replacement therapy (HRT) is a treatment that replaces the estrogen and progesterone no longer produced by the body after menopause and is used to relieve the symptoms of menopause, such as hot flushes and night sweats [3]. It might improve the quality of life of women with menopausal symptoms and risk factors related to cardiovascular disease or osteoporosis [3,4]. However, it introduces small risks of diseases such as breast cancer, cardiovascular disease, stroke, and venous thromboembolism (VTE) depending on the type of HRT, HRT duration, and individual health risks [3,4].

All histologic subtypes of epithelial ovarian cancer (EOC), which account for about 90% of ovarian cancer cases, are associated with the expressions of estrogen and progesterone receptors, and the differential expressions of these hormone receptors influence survival, especially in high-grade serous and endometrioid carcinoma [5,6]. However, meta- and pooled analyses showed postmenopausal estrogen-only therapy (ET) and estrogen-progesterone therapy (EPT) are associated with an increased risk of ovarian cancer [7,8,9]. More specifically, the risk of ovarian cancer was higher in current or recent users and for serous and endometrioid subtypes and increased with HRT duration, especially for durations of ≥10 years [7,9]. On the other hand, a large-scale international study found that the use of ET or EPT for at least five years prior to EOC diagnosis was associated with better survival [10].

Effects of HRT after EOC diagnosis on survival have been investigated several times [11,12,13,14,15,16,17,18]. Four retrospective studies and two randomized controlled trials (RCTs) concluded postoperative HRT did not have a significant effect on EOC survival [11,12,13,14,15,16]. However, a prospective study and an RCT found HRT use after diagnosis of EOC and postoperative HRT for EOC, respectively, were associated with better survival [17,18]. Moreover, a recent meta-analysis of these three RCTs concluded postoperative HRT might slightly improve survival [19]. However, this result should be interpreted with caution because of poor quality evidence, small sample sizes, premature discontinuation, and heterogeneous populations.

Thus, although previous studies have not shown negative effects on survival, the effect of postoperative HRT in women with ovarian cancer remains unclear. Well-designed large-scale trials are needed to determine the safety and benefits of postoperative HRT on survival in women with ovarian cancer. This study was undertaken to determine the impact of postoperative HRT on survival in women with ovarian cancer using Korean Health Insurance Review and Assessment Service (HIRA) data.

## 2. Materials and Methods

South Korea has a universal health insurance system (National Health Insurance), which covers approximately 98% of the population [20]. The HIRA claims data represent the claims made by 23 million women per year [20]. In the present study, we used the claims data of women with diagnostic codes for ovarian cancer first registered in the HIRA between 1 July 2007 and 31 December 2018 and followed their subsequent claims up to 30 June 2020. The HIRA dataset uses anonymous identification codes to protect personal information in accordance with the Korean Bioethics and Safety Act, and thus, the requirement for study approval was waived by the Institutional Review Board of Inha University Hospital (No. 2021-02-012) on 19 April 2022. Informed consent was not required.

The codes used to select eligible patients were based on the 10th revision of the International Statistical Classification of Diseases and Related Health Problems (ICD-10), Health Insurance Medical Care Expenses (2017 and 2018 version), and HIRA Drug Ingredient Codes. Using the Korea Central Cancer Registry as a reference, women with ovarian cancer were defined to have five or more diagnostic codes for ovarian cancer (ICD-10: C56x) and V193 and no diagnostic code for any other cancer within the 2-year period preceding the first diagnostic code entry for ovarian cancer. The V code is a special code for women with any ICD-10 cancer code in South Korea and was established by the Korean Ministry of Health and Welfare in 2008. Women with C56x codes registered between 1 July 2007 and 31 December 2008 were excluded to ensure the exclusive inclusion of women with newly diagnosed ovarian cancer.

The following exclusion criteria were applied: an age of >60 when the first diagnostic code was registered; non-receipt of surgery as a primary treatment; non-receipt of surgery within 60 days before the first diagnostic code for ovarian cancer was registered; a diagnosis of VTE before primary surgery for ovarian cancer; receipt of HRT before primary surgery; receipt of HRT first at >one year after primary surgery; receipt of HRT for <30 days during the year following primary surgery and ≥30 days later than one year after primary surgery; and women of an age < 19 years old when the first diagnostic code for ovarian cancer was registered. Postoperative VTE was defined as the receipt of prescriptions for anticoagulants more than twice simultaneously with diagnostic codes for VTE (ICD-10: I80.2, I80.3, I26.0, I26.9) after primary surgery. Women in the HRT group had received prescriptions for HRT ≥ 30 days during the year following primary surgery, and women in the control group did not receive a prescription for HRT or received an HRT prescription for <30 days during the year following primary surgery and for <30 days later than one year after primary surgery.

Low socioeconomic status (SES) was defined as the use of Medicaid. Charlson comorbidity indices (CCIs) were calculated using data obtained during the year before ovarian cancer was registered with the HIRA, as described by Quan [21]. Primary surgery was defined using surgery codes for salpingo-oophorectomy (bilateral or unilateral) or ovarian cystectomy and/or total hysterectomy and a concurrent diagnostic code for ovarian cancer or those surgery codes within 60 days before the first diagnostic code for ovarian cancer. When two or more surgeries were performed, the first surgery was defined as the primary surgery. Other surgeries were defined as additional surgeries implemented simultaneously with primary surgery. Neoadjuvant chemotherapy was defined as chemotherapy during the 12 weeks preceding primary surgery, and adjuvant chemotherapy was defined as chemotherapy within 12 weeks after primary surgery. Chemotherapy was defined using prescription codes for chemotherapy (carboplatin, cisplatin, cyclophosphamide, docetaxel, fluorouracil, gemcitabine, ifosfamide, irinotecan, liposomal doxorubicin, mitomycin, paclitaxel, topotecan, bevacizumab) with a concurrent diagnostic code for ovarian cancer. Hormone therapy was defined as prescription codes for hormone therapy through oral and transdermal routes (estrogen therapy (ET), estrogen-progesterone therapy (EPT), tibolone) with a concurrent diagnostic code for ovarian cancer. Use of prophylactic anticoagulants for postoperative VTE was defined as the receipt of anticoagulants more than twice without diagnostic codes for VTE after primary surgery. Anticoagulants included unfractionated heparin, low-molecular-weight heparin, fondaparinux, warfarin, and direct oral anticoagulants. Overall survival was defined as time from first diagnostic code entry for ovarian cancer to death or expected death. For women with no claim data for >one year, expected death date was defined by adding a random date (from 1 to 365 days) to the last medical record date.

### Statistical Analyses

The analysis was performed using SAS^®^ Enterprise Guide^®^ version 7.15 (SAS Institute, Inc., Cary, NC, USA) and R versions 3.5.1 and 4.1.1 (R Foundation for Statistical Computing, Vienna, Austria). Continuous variables were analyzed using the two-sample t-test, and categorical variables using the chi-square test or Fisher’s exact test. Univariable analysis was performed using the Cox proportional hazards regression model to investigate associations between variables and overall survival (OS) and between HRT duration and OS. Variables found to be significantly associated with OS by univariable analysis were assessed using the proportional hazards assumption. Multivariable analysis was performed using the stratified Cox proportional hazards model. Variables that did not satisfy the proportional hazards assumption were used as stratification variables and excluded from the model, and variables that satisfied the proportional hazards assumption were used as covariables. The proportional hazards assumption was examined using Schoenfeld residuals over time and a test developed by Grambsch and Therneau [22]. For variables with a small event size, the Firth penalized maximum-likelihood estimation was applied to reduce bias in 95% confidence intervals (CIs) and parameter estimates [23]. Kaplan–Meier curves were constructed using the log-rank test to examine the cumulative probability of OS. Trends of restricted mean survival times (RMSTs) and postoperative VTE incidence rates were analyzed using the Jonckheere–Terpstra test. All tests were two-sided, and *p* values of <0.05 were considered statistically significant. The mean imputation method was used to account for missing values.

## 3. Results

Initially, the data of 29,482 women with five or more diagnostic codes for ovarian cancer first registered by the HIRA between 2007 and 2018 were extracted. Of these, 1784 women met the study eligibility criteria (Figure 1); that is, 263 (14.7%) women received HRT (the HRT group) and 1521 (85.3%) did not (the control group).

### 3.1. Characteristics of Women with Ovarian Cancer according to Receipt of HRT

The baseline characteristics of the 1784 study subjects are shown in Table 1, and the age distributions of the HRT and control groups are shown in Appendix A. For all study subjects, the mean follow-up was 5.6 ± 2.9 years (6.4 ± 2.9 years in the HRT group and 5.5 ± 2.9 years in the control group). HRT was started at a mean of 127.2 ± 93.7 days after primary surgery, and the estimated mean duration of HRT was 3.48 ± 2.91 years; 25.9% of the 263 women in the HRT group received HRT for >5 years.

### 3.2. OS Risk Factors in Women with Ovarian Cancer

After adjustment for other variables, receipt of HRT was significantly associated with a longer OS (hazard ratio (HR) 0.618; 95% CI 0.414–0.922; *p* = 0.018). However, OS decreased significantly with age, the presence of postoperative VTE, and receipt of other surgery or prophylactic anticoagulants (age: HR 1.017; 95% CI 1.005–1.030; *p* = 0.006) (other surgery: HR 1.571; 95% CI 1.070–2.306; *p* = 0.021) (postoperative VTE: HR 5.522; 95% CI 3.540–8.613; *p* < 0.001) (prophylactic anticoagulants: HR 2.671; 95% CI 1.862–3.831; *p* < 0.001). CCIs and primary surgery methods were not found to be risk factors of OS (Table 2).

In addition, multivariate analysis adjusted for other variables showed 5-year OS was significantly greater in the HRT group (HR 0.583; 95% CI 0.359–0.947; *p* = 0.029) and in women with a middle or high SES than women with a low SES (HR 0.367; 95% CI 0.195–0.691; *p* = 0.002). Furthermore, 5-year OS decreased significantly with age, the presence of postoperative VTE, and receipt of prophylactic anticoagulants (age: HR 1.018; 95% CI 1.004–1.033; *p* = 0.01) (postoperative VTE: HR 5.491; 95% CI 3.388–8.901; *p* < 0.001) (prophylactic anticoagulants: HR 2.656; 95% CI 1.795–3.928; *p* < 0.001). CCIs, another surgery, and primary surgery methods were not found to be risk factors for OS (Appendix A).

### 3.3. OS and RMST according to Receipt of HRT

Over a follow-up of 11.5 years, 246 of the 1784 study subjects (13.8%) (27 in the HRT group (10.3%) and 219 women in the control group (14.4%)) died, and during a follow-up of 5 years, 187 women (10.5%) (18 women in the HRT group (6.8%) and 169 women in the control group (11.1%)) died. Median OS could not be calculated because the death rate was less than 50%. A comparison of Kaplan–Meier curves showed OS was significantly greater in the HRT group (*p* = 0.016) (Figure 2A); OS was 85.3% (95% CI 78.7–89.9%) in the HRT group and 76.6% (95% CI 72.6–80.2%) in the control group. Five-year OS in the HRT and control group was 95% and 87.2%, respectively (95% CI 88.0–95.1% and 95% CI 85.2–88.9%).

RMST analysis showed the ratio of women that received HRT to women that did not increased significantly with time (*p* < 0.001 for OS and *p* = 0.017 for 5-year OS). RMSTs at 1 and 2 years after diagnosis of ovarian cancer were not significant in the two study groups, but RMSTs at 3, 4, 5, 10, and 11.5 years were significantly greater for women in the HRT group (difference of RMST: 3 years, 0.052 (95% CI 0.006–0.098), *p* = 0.026; 4 years, 0.099 (95% CI 0.026–0.173), *p* = 0.008; 5 years, 0.157 (95% CI 0.054–0.261), *p* = 0.003; 10 years, 0.441 (95% CI 0.127–0.754), *p* = 0.005; 11.5 years, 0.562 (95% CI 0.167–0.957), *p* = 0.005) (Table 3).

The OS of women that received surgery alone, surgery and adjuvant chemotherapy, platinum-based chemotherapy, or other chemotherapeutic agents were not significantly different in the HRT and control groups (Appendix A).

### 3.4. OS and RMSTs according to HRT Duration in Women with Ovarian Cancer

Kaplan–Meier curves showed OS was significantly associated with HRT duration (*p* < 0.001) (Figure 2B). OS was significantly higher for women that received HRT for ≥5 years than for women that received HRT for <0.5 years (HR 0.234; 95% CI 0.059–0.936; *p* = 0.040). The OS of women that received HRT for <0.5 years or for other periods were not significantly different (0.5–1 years vs. <0.5 years: HR 2.916; 95% CI 0.954–8.915; *p* = 0.061) (1–2 years vs <0.5 years: HR 1.284; 95% CI 0.419–3.932; *p* = 0.661) (2–3 years vs. <0.5 years: HR 0.674; 95% CI 0.143–3.180; *p* = 0.619) (3-4 years vs. <0.5 year: HR 0.518; 95% CI 0.110–2.443; *p* = 0.406) (4–5 years vs. <0.5 year: HR 0.094; 95% CI 0.004–2.011; *p* = 0.131). RMSTs for OS at 1 and 2 years were not significantly related to HRT duration, whereas those at 3, 4, 5, 10, and 11.5 years showed significant HRT-dependent increases (3 years, *p* = 0.007; 4 years, *p* = 0.016; 5 years, *p* = 0.011; 10 years, *p* = 0.03; 11.5 years, *p* = 0.03) (Table 3).

## 4. Discussion

In the present study, multivariable analysis adjusted for covariables showed OS was significantly greater in women that received HRT than in those that did not. Furthermore, RMSTs for OS showed the ratio of women with HRT to women without HRT significantly increased with time, especially from 3 years after diagnosis of ovarian cancer, and duration of HRT was associated with a significant OS increase in women that received HRT for >5 years as compared with women that received HRT for ≤0.5 years. Furthermore, RMSTs at 3, 4, 5, 10, and 11.5 years significantly increased with HRT duration. Despite the limitations associated with the use of claims data, this study demonstrates the impact of postoperative HRT on the survival of women diagnosed with ovarian cancer at ≤60 years of age in the largest cohort studied to date.

Women with ovarian cancer have a median age at diagnosis of 57–63 years [2,24,25,26,27,28], though in our study, mean ages of women in the HRT and control groups were 41.5 ± 8.5 and 41.0 ± 11.4 years, respectively. This apparent discrepancy might be explained by the following. First, we intended to evaluate the effects of HRT initiated after primary surgery for ovarian cancer. Therefore, women diagnosed with ovarian cancer after 60 years of age were excluded because most contemporary guidelines recommend HRT be initiated before 60 years of age based on considerations of the benefits of HRT and associated risks of cardiovascular disease, stroke, and VTE [3,4]. Second, although the HIRA dataset does not provide information about histologic types, we attempted to include mainly EOC cases. Therefore, we excluded women diagnosed with ovarian cancer before 19 years of age to minimize non-EOC cases. Typically, women with ovarian cancer that present with advanced-stage disease receive combined cytoreductive surgery and adjuvant chemotherapy [1,29]. Moreover, when upfront surgery is contraindicated for medical reasons or when optimal cytoreduction cannot be achieved, neoadjuvant chemotherapy prior to cytoreductive surgery and adjuvant chemotherapy provides an alternative option [30]. However, more than 80% of women included in our study received “BSO, USO, or ovarian cystectomy”, or “surgery alone”, which suggests a high percentage of women with early-stage disease. Thus, high percentages of young women and women with early-stage disease probably explain the higher survival rates observed in our cohort [24,25,26,27,28].

Postoperative HRT might be a safe, therapeutic modality in terms of survival in women with ovarian cancer [11,12,13,14,15,16,17,18], but the survival benefit of postoperative HRT is unclear [19]. All three RCTs that evaluated the effect of postoperative HRT on survival in ovarian cancer contained only 31 to 75 participants that received HRT, enrolled participants without considering menopausal symptoms, and reported low compliances [15,16,18]. In the one RCT that reported postoperative HRT improved survival, the median estimated duration of HRT was only 1.14 years because of low compliance, and the median follow-up period was 19.1 years. In this RCT, differences between the OS of women that did or did not receive HRT also increased with time (RMST at 20 years: 8.5 years for women that received HRT and 5.7 years for women that did not; an absolute difference of 2.8 years (95% CI, 0.3–5.2 years)) [18]. In our study, postoperative HRT improved survival when administered for 3.48 ± 2.91 years. Furthermore, women with HRT were followed for longer than women without HRT (6.4 ± 2.9 vs. 5.5 ± 2.9 years), which suggested longer survival for women with HRT. In addition, the difference between OS in the HRT group and the control group increased with time, especially from 3 years. However, differences in survival were minimal (RMST at 11.5 years: 10.5 years in the HRT group and 9.9 years in the control group; an absolute difference of 0.6 years (95% CI, 0.2–1.0 years)). Moreover, we found survival was further improved by long-term HRT (>5 years) and relatively long-term HRT (from 3 years after diagnosis of ovarian cancer). These results support the long-term use of HRT.

The reduced survival of women that received prophylactic anticoagulants observed in the present study might be explained by an 11.6-fold higher incidence rate of postoperative VTE in women administered prophylactic anticoagulants (Appendix A). In addition, these women had a shorter mean time between primary surgery and VTE diagnosis, were older, had a higher CCI, and higher frequencies of total hysterectomy, other surgeries, adjuvant chemotherapy, neoadjuvant chemotherapy, and chemotherapy (Appendix A). Notably, the incidence rates of postoperative VTE significantly decreased with time in women administered prophylactic anticoagulants (*p* = 0.012), but no such trend was evident in women not administered prophylactic anticoagulants (Appendix A).

This nationwide, population-based cohort study is the largest conducted to date to investigate the impact of postoperative HRT on survival in ovarian cancer. The limitations of this study are mainly associated with the use of claims data. First, diseases and treatments were defined using diagnostic and prescription codes without reviewing medical records, and thus, a few women with incorrect codes may have been misdefined. Second, women not administered HRT included a few women (5.1%) that received HRT for a short time, as women that received HRT for <30 days during the year after primary surgery and later than one year after primary surgery were classified as women that did not receive HRT. However, in our opinion, this classification allowed more accurate access to the impact of HRT on survival. Finally, we could not evaluate relationships between HRT and BMI, stage or histologic type of ovarian cancer, residual tumors, number of chemotherapy cycles or BRCA mutational status because the HIRA dataset did not provide this information.

## 5. Conclusions

This retrospective, nationwide analysis of claims data shows postoperative HRT has survival benefits in women with ovarian cancer, and that the impact of HRT on survival increases with time and HRT duration. Our results indicate postoperative HRT has potential use as a maintenance therapy in women diagnosed at ≤60 years with ovarian cancer. We recommend a large-scale RCT be undertaken to confirm our results.

## Figures and Tables

**Figure 1 cancers-14-03090-f001:**
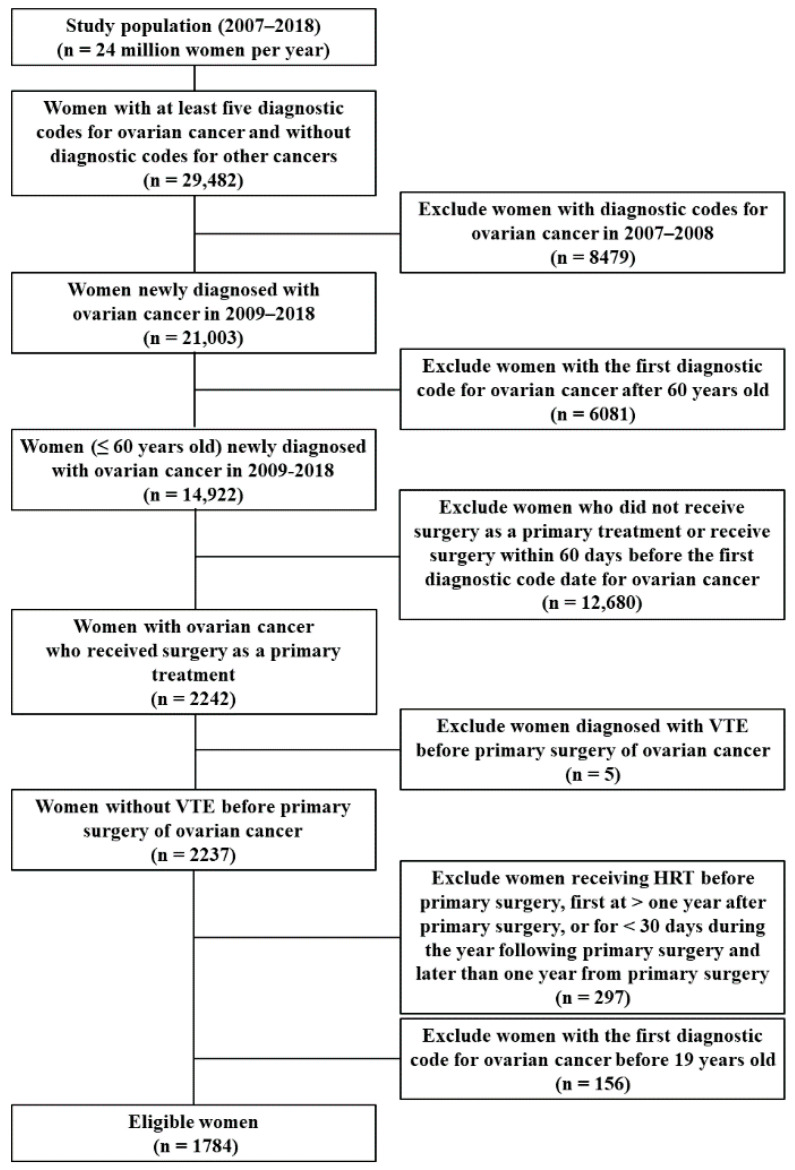
Flow chart of study subject selection.

**Figure 2 cancers-14-03090-f002:**
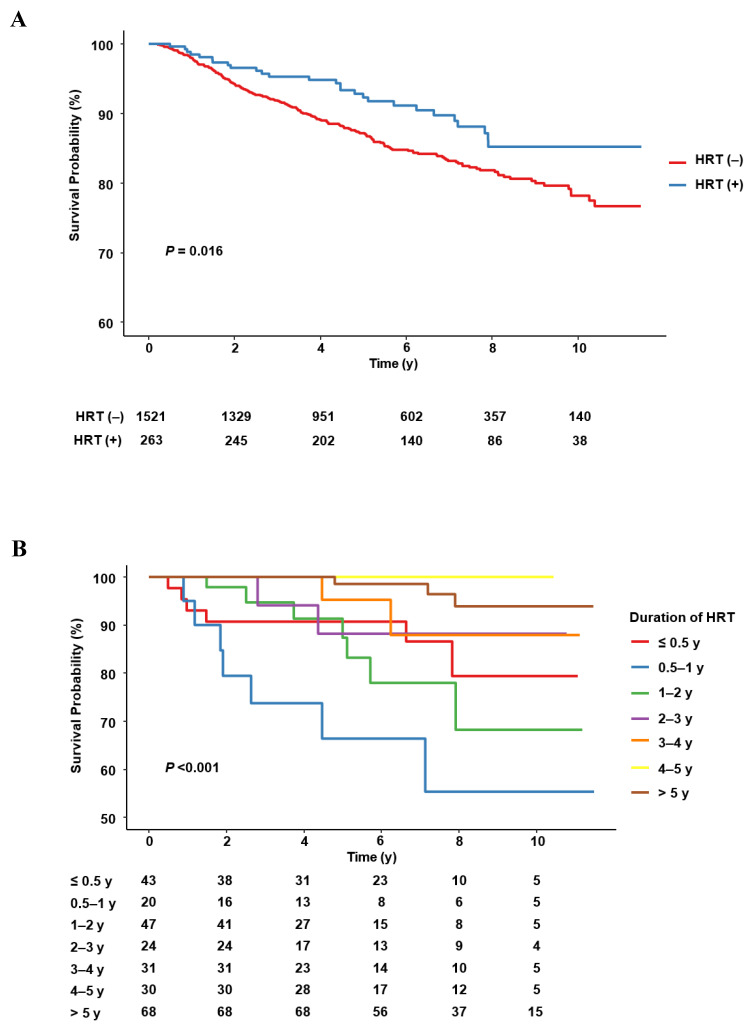
OS according to HRT use and duration in women with ovarian cancer. (**A**) OS according to HRT, (**B**) OS according to HRT duration.

**Table 1 cancers-14-03090-t001:** Characteristics of women with ovarian cancer according to HRT administration.

	No. (%)			
Characteristics	Total	HRT (−)	HRT (+)	*p* Value
No. of women	1784 (100)	1521 (85.3)	263 (14.7)	
Age, mean (SD), year	41.0 (11.0)	41.0 (11.4)	41.5 (8.5)	0.441
SES				
Middle or high SES	1743 (97.7)	1484 (97.6)	259 (98.5)	0.362
Low SES	41 (2.3)	37 (2.4)	4 (1.5)	
CCI				
0	943 (52.9)	807 (53.1)	136 (51.7)	0.988
1	549 (30.8)	466 (30.6)	83 (31.6)	
2	192 (10.8)	164 (10.8)	28 (10.6)	
3	59 (3.3)	50 (3.3)	9 (3.4)	
≥4	41 (2.3)	34 (2.2)	7 (2.7)	
Year of cancer diagnosis				
2009	141 (7.9)	117 (7.7)	24 (9.1)	0.004
2010	168 (9.4)	133 (8.7)	35 (13.3)	
2011	143 (8.0)	114 (7.5)	29 (11.0)	
2012	173 (9.7)	150 (9.9)	23 (8.8)	
2013	157 (8.8)	129 (8.5)	28 (10.6)	
2014	200 (11.2)	166 (10.9)	34 (12.9)	
2015	193 (10.8)	164 (10.8)	29 (11.0)	
2016	189 (10.6)	167 (11.0)	22 (8.4)	
2017	205 (11.5)	190 (12.5)	15 (5.7)	
2018	215 (12.1)	191 (12.6)	24 (9.1)	
Methods of primary surgery				
BSO, USO, or ovarian cystectomy	1542 (86.4)	1314 (86.4)	228 (86.7)	0.895
Total hysterectomy ± BSO, USO, or ovarian cystectomy	242 (13.6)	207 (13.6)	35 (13.3)	0.895
Other surgeries ^a^	290 (16.3)	248 (16.3)	42 (16.0)	0.892
Types of primary surgery				
Surgery alone	1441 (80.8)	1217 (80.0)	224 (85.2)	0.05
Surgery + adjuvant chemotherapy	337 (18.9)	298 (19.6)	39 (14.8)	0.068
Neoadjuvant chemotherapy + surgery ± adjuvant chemotherapy	6 (0.3)	6 (0.4)	0 (0.0)	0.601 ^b^
Chemotherapy				
Platinum-based chemotherapy	335 (18.8)	298 (19.6)	37 (14.1)	0.034
Other agents	281 (15.8)	249 (16.4)	32 (12.2)	0.084
Bevacizumab ± any agents	7 (0.4)	6 (0.4)	1 (0.4)	>0.999 ^b^
Postoperative VTE				
(−)	1744 (97.8)	1489 (97.9)	255 (97.0)	0.343
(+)	40 (2.2)	32 (2.1)	8 (3.0)	
Prophylactic anticoagulants				
(−)	1652 (92.6)	1405 (92.4)	247 (93.9)	0.377
(+)	132 (7.4)	116 (7.6)	16 (6.1)	
HRT				
Estrogen	196 (11.0)	47 (3.1)	149 (56.7)	<0.001
Estrogen + Progesterone	151 (8.5)	83 (5.5)	68 (25.9)	<0.001
Tibolone	161 (9.0)	65 (4.3)	96 (36.5)	<0.001
Duration of HRT, year				
≤0.5			43 (16.3)	
0.5–1			20 (7.6)	
1–2			47 (17.9)	
2–3			24 (9.1)	
3–4			31 (11.8)	
4–5			30 (11.4)	
>5			68 (25.9)	
Time between primary surgery and HRT use, mean (SD), d			127.2 (93.7)	

Abbreviations: BSO, bilateral salpingo-oophorectomy; CCI, Charlson comorbidity index; HRT: hormone replacement therapy; SD: standard deviation; SES, socioeconomic status; USO, unilateral salpingo-oophorectomy; VTE, venous thromboembolism. ^a^ Appendectomy, bowel resection, cholecystectomy, end-to-end ureteroureterostomy, pancreatectomy, partial gastrectomy, partial hepatectomy, pelvic and/or para-aortic lymph node dissection, splenectomy, stripping of other peritoneal surfaces, stripping of the diaphragm, ureteroneocystostomy. ^b^ Fisher’s exact test was used for this analysis.

**Table 2 cancers-14-03090-t002:** Associations between risk factors and OS.

	Univariable Analysis		Multivariable Analysis ^a,b^	
Variable	HR (95% CI)	*p* Value	HR (95% CI)	*p* Value
Age, year	1.026 (1.014–1.039)	<0.001	1.017 (1.005–1.030)	0.006
SES				
Low SES	ref	<0.001		
Middle or high SES	0.314 (0.179–0.549)			
CCI				
0	ref			
1	0.847 (0.631–1.137)	0.269		
2	0.953 (0.632–1.440)	0.821		
3	0.989 (0.485–2.016)	0.975		
≥4	0.936 (0.413–2.118)	0.873		
Methods of primary surgery				
BSO, USO, or ovarian cystectomy	ref	0.027	ref	0.328
Total hysterectomy ± BSO, USO, or ovarian cystectomy	1.466 (1.044–2.058)		0.796 (0.504–1.257)	
Other surgeries ^c^				
(−)	ref	<0.001	ref	0.021
(+)	1.690 (1.265–2.257)		1.571 (1.070–2.306)	
Types of primary surgery				
Surgery alone	ref			
Surgery + adjuvant chemotherapy	2.101 (1.506–2.932)	<0.001 ^b^		
Neoadjuvant chemotherapy + surgery ± adjuvant chemotherapy	1.373 (0.084–22.43)	0.824 ^b^		
Postoperative VTE				
(−)	ref	<0.001	ref	<0.001
(+)	7.731 (5.061–11.810)		5.522 (3.540–8.613)	
Prophylactic anticoagulants				
(−)	ref	<0.001	ref	<0.001
(+)	4.139 (2.985–5.739)		2.671 (1.862–3.831)	
HRT				
(−)	ref	0.017	ref	0.018
(+)	0.615 (0.412–0.917)		0.618 (0.414–0.922)	

Abbreviations: BSO, bilateral salpingo-oophorectomy; CCI, Charlson comorbidity index; CI, confidence interval; HR, hazard ratio; HRT: hormone replacement therapy; SES, socioeconomic status; USO, unilateral salpingo-oophorectomy; VTE, venous thromboembolism. ^a^ Stratified Cox proportional hazards model was used for multivariable model. ^b^ Firth penalized maximum-likelihood estimation was applied for this analysis. ^c^ Appendectomy, bowel resection, cholecystectomy, end-to-end ureteroureterostomy, pancreatectomy, partial gastrectomy, partial hepatectomy, pelvic and/or para-aortic lymph node dissection, splenectomy, stripping of other peritoneal surfaces, stripping of the diaphragm, ureteroneocystostomy.

**Table 3 cancers-14-03090-t003:** RMSTs according to HRT use and duration in women with ovarian cancer.

	1 Year		2 Year		3 Year		4 Year		5 Year		10 Year		11.5 Year	
	RMST (95% CI)	*p* Value	RMST (95% CI)	*p* Value	RMST (95% CI)	*p* Value	RMST (95% CI)	*p* Value	RMST (95% CI)	*p* Value	RMST (95% CI)	*p* Value	RMST (95% CI)	*p* Value
All women														
HRT (−)	0.993 (0.990–0.996)		1.954 (1.943–1.965)		2.883 (2.860–2.905)		3.788 (3.752–3.823)		4.669 (4.618–4.719)		8.792 (8.643–8.940)		9.919 (9.732–10.105)	
HRT (+)	0.997 (0.993–1.001)		1.974 (1.954–1.993)		2.935 (2.895–2.975)		3.887 (3.823–3.951)		4.826 (4.736–4.916)		9.233 (8.957–9.509)		10.481 (10.132–10.829)	
Difference	0.004 (−0.001–0.009)	0.095	0.02 (−0.003–0.042)	0.086	0.052 (0.006–0.098)	0.026	0.099 (0.026–0.173)	0.008	0.157 (0.054–0.261)	0.003	0.441 (0.127–0.754)	0.006	0.562 (0.167–0.957)	0.005
Ratio	1.004 (0.999–1.009)	0.095	1.01 (0.999–1.022)	0.085	1.018 (1.002–1.034)	0.026	1.026 (1.007–1.046)	0.008	1.034 (1.012–1.056)	0.003	1.050 (1.015–1.087)	0.005	1.057 (1.017–1.098)	0.005
Duration of HRT, y														
≤0.5	0.984 (0.961–1.008)		1.902 (1.806–1.999)		2.809 (2.629–2.990)		3.716 (3.450–3.983)		4.623 (4.270–4.4.976)		8.864 (8.013–9.714)		9.716 (8.714–10.719)	
0.5–1	0.995 (0.985–1.005)		1.891 (1.764–2.018)		2.681 (2.408–2.954)		3.402 (2.934–3.870)		4.100 (3.447–4.754)		7.101 (5.397–8.805)		7.695 (5.730–9.660)	
1–2	1.000 (1.000–1.000)		1.989 (1.968–2.010)		2.952 (2.884–3.021)		3.891 (3.754–4.027)		4.804 (4.584–5.024)		8.542 (7.601–9.483)		9.346 (8.165–10.528)	
2–3	1.000 (1.000–1.000)		2.000 (2.000–2.000)		2.989 (2.967–3.010)		3.930 (3.796–4.063)		4.834 (4.583–5.085)		9.245 (8.255–10.236)		9.930 (8.822–11.039)	
3–4	1.000 (1.000–1.000)		2.000 (2.000–2.000)		3.000 (3.000–3.000)		4.000 (4.000–4.000)		4.975 (4.926–5.023)		9.461 (8.756–10.166)		10.143 (9.315–10.972)	
4–5	1.000 (1.000–1.000)		2.000 (2.000–2.000)		3.000 (3.000–3.000)		4.000 (4.000–4.000)		5.000 (5.000–5.000)		10.000 (10.000–10.000)		10.442 (10.442–10.442)	
>5	1.000 (1.000–1.000)		2.000 (2.000–2.000)		3.000 (3.000–3.000)		4.000 (4.000–4.000)		4.997 (4.991–5.003)		9.811 (9.599–10.022)		10.226 (9.985–10.466)	
Trend		0.099		0.051		0.007		0.016		0.011		0.030		0.030

Abbreviations: RMST, restricted mean survival times.

## Data Availability

The data that support the findings of this study are available from the Health Insurance Review and Assessment Service (HIRA), but restrictions apply to the availability of these data which were used under license for the current study, and thus are not publicly available. Data are, however, available from the authors upon reasonable request and with permission of the HIRA.

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
