# Peer review of "Postoperative Hormone Replacement Therapy and Survival in Women with Ovarian Cancer"

_cancers, 2022, doi:10.3390/cancers14133090_

Round 1

Reviewer 1 Report

  1. The difference in the size of the HRT (-) and HRT (+) groups is noticeable.
  2. When analyzing parameters such as overall survival among patients with ovarian cancer, it is necessary to take into account the basic prognostic factors, i.e. the degree of FIGO and the surgical outcome (optimal, non-optinal). In my opinion, the lack of these data does not give certainty that both groups were homogeneous and can be compared.
  3. Taking into account the types of surgeries, it can be assumed that 86 patients were in the I FIGO stage. Was it so?
  4. What were the histological types of ovarian cancer in both groups?
    Did the lack of adjuvant treatment mean that most patients had low-grade ovarian cancer (formerly described as G1)?
  5. There is no data on relapse of disease in the described groups. Was HRT administered during relapse treatment and after that time?
  6. It is difficult to accept the statement that the type of surgery is a risk factor associated with a shorter OS (Table 2) More radical surgery indicates a more advanced disease, which determines shorter survival
  7. Table 1 shows that the study group included patients after unilateral adnexectomy (USO) or ovarian cystectomy. Do the authors consider cystectomy a sufficient treatment for ovarian cancer? Was the staging performed recommended by international scientific societies?
  8. Were HRT also administered in premenopausal patients after USO?
  9. The summary of the discussion should mention the strengths and weaknesses of the work
  10. With such material, it might be better to write a paper on HRT results in a group of women with early ovarian cancer, e.g. after fertility-sparing surgery.

Reviewer 2 Report

In this paper, the authors summarize the impact of postoperative HRT on survival in women with ovarian cancer using Korean Health Insurance Review & Assessment Service (HIRA) data. The manuscript is straightforward, well written, and concise and has clear results, within the scope of a retrospective analysis. Definitely deserves to be published and is a valuable contribution to the “cancersjournal. Some minor comments need to be addressed before publication.

[1] 4. Discussion”, Page 10 of 13, Lines 255-257:

Typically, women with ovarian cancer that present with advanced-stage disease receive combined cytoreductive surgery and adjuvant chemotherapy [29,30].”.

At that point, the authors should report that three or more cycles of neoadjuvant chemotherapy (NACT) prior to debulking surgery and adjuvant chemotherapy is an alternative option for selected patients. However, there is currently a lack of consensus about who are the best candidates for this strategy, and how to best select them. Importantly, NACT offers the opportunity to test upfront chemosensitivity and to identify patients at higher risk of relapse.

Recommended reference: Moschetta M, et al. Neoadjuvant treatment for newly diagnosed advanced ovarian cancer: where do we stand and where are we going? Ann Transl Med. 2020;8(24):1710.

[2] “4. Discussion”, Page 11 of 13, Lines 292-293:

The limitations of this study are mainly associated with the use of claims data.”.

The authors should also incorporate the absence of information related to the BRCA mutational status within the limitations of the study.

Reviewer 3 Report

Nicely written and data are presented well.

Author Response

Thank you very much for your kind comment.

Reviewer 4 Report

The manuscript “Postoperative hormone replacement therapy and survival in women with ovarian cancer” by Eunjeong Ji and co-authors to investigate the impact of postoperative HRT on survival in women with ovarian cancer using the nationwide cohort study. Women aged ≤ 60 diagnosed with ovarian cancer and received primary surgery were followed-up for 5.6 ± 2.9 years. Mean ages of women administered HRT (the HRT group; n=263) or not administered HRT (the control group; n=1,521) were 41.5 ± 8.5 and 41.0 ± 11.4 years, respectively. After adjustment for co-variables, OS was significantly greater in the HRT group (HR 0.618; 95% CI 0.414-0.922; P = 0.018). Kaplan–Meier curve analysis showed OS was significantly higher in the HRT group (85.3% vs. 76.6%; P = 0.016). The ratio of women with HRT to women without HRT increased significantly with time (restricted mean survival times for OS, P <0.001). In addition, OS was significantly greater for those that received HRT for > 5 years than for those that received HRT for ≤ 0.5 years (HR 0.234; 95% CI 0.059-0.936; P = 0.040). Postoperative HRT improved survival among women with ovarian cancer. The impact of HRT on survival increased with time and treatment duration. Some concerns which must be taken into account before the work can be reconsidered for publication.

Comment

Do author analysis the association between HRT treatment and clinical parameters, ex: tumor type, stage? 

Reviewer 5 Report

Ji et al. investigated the impact of HRT treatment on the survival of women who underwent surgery due to ovarian cancer. The study focused on women who were diagnosed with cancer when they were <60 years of age. Other specific inclusion criteria were described in the methods section. Overall, the study is well designed and well written. Findings have clinical significance. The manuscript might benefit from consideration of the issues listed below:

1) The Introduction section could improve with a clear definition of HRT and its harms and benefits.

2) What are the specifics of the HRT treatment that the participants received (i.e., dose, route, frequency etc)?

3) Osteoporosis, minimal trauma fracture, and increased hospitalisation are common in post-menopausal women. Did the authors account for the differences between the HRT and non-HRT groups in terms of these secondary outcomes that can have a direct impact on the overall survival?

4) Extending from the previous comment, are there any data from the DEXA scan that compares these two groups of participants?

5) Did the authors observe increased CVD in the HRT group?

Round 2

Reviewer 1 Report

The explanations are sufficient. The manuscript has been significantly improved

Reviewer 4 Report

The revised manuscript “Postoperative hormone replacement therapy and survival in women with ovarian cancer” have adequately addressed my previous concerns and the paper is now acceptable for publication.